# Triterpene Acids of Loquat Leaf Improve Inflammation in Cigarette Smoking Induced COPD by Regulating AMPK/Nrf2 and NFκB Pathways

**DOI:** 10.3390/nu12030657

**Published:** 2020-02-28

**Authors:** Tunyu Jian, Xiaoqin Ding, Jiawei Li, Yuexian Wu, Bingru Ren, Jing Li, Han Lv, Jian Chen, Weilin Li

**Affiliations:** 1Institute of Botany, Jiangsu Province and Chinese Academy of Sciences, Nanjing 210014, China; jiantunyu1986@163.com (T.J.); dingxiao_qin@126.com (X.D.); lijiawei36@foxmail.com (J.L.); wuyuexian1993@163.com (Y.W.); Bingruren@126.com (B.R.); chenjian80@aliyun.com (J.C.); 2Department of Food Science and Technology, College of Light Industry and Food Engineering, Nanjing Forestry University, Nanjing 210037, China; lijing1020@njfu.edu.cn; 3Co-Innovation Center for Sustainable Forestry in Southern China, Forestry College, Nanjing Forestry University, Nanjing 210037, China; lwlcnbg@cnbg.net

**Keywords:** loquat leaf, triterpene acids, chronic obstructive pulmonary disease, cigarette smoke, inflammation, AMPK, Nrf2, NFκB, iNOS

## Abstract

Cigarette smoking (CS) is believed to be an important inducement in the pathological development of chronic obstructive pulmonary disease (COPD), a progressive lung disease. Loquat is an Asian evergreen tree commonly cultivated for its fruit. Its leaf has long been used as an important material for both functional and medicinal applications in the treatment of lung disease in China and Japan. As the principal functional components of loquat leaf, triterpene acids (TAs) have shown notable anti-inflammatory activity. However, their protective activity and underlying action of mechanism on CS-induced COPD inflammation are not yet well understood. In the present study, male C57BL/6 mice were challenged with CS for 12 weeks, and from the seventh week of CS exposure, mice were fed with TAs (50 and 100 mg/kg) for 6 weeks to figure out the therapeutic effect and molecular mechanism of TAs in CS-induced COPD inflammation. The results demonstrate that TA suppressed the lung histological changes in CS-exposed mice, as evidenced by the diminished generation of pro-inflammatory cytokines, including interleukin 1β (IL-1β), IL-2, IL-6, and tumor necrosis factor α (TNF-α). Moreover, TA treatment significantly inhibited the malondialdehyde (MDA) level and increased superoxide dismutase (SOD) activity. In addition, TAs increased the phosphorylation of AMP-activated protein kinase (AMPK) and nuclear factor erythroid-2-related factor-2 (Nrf2) expression level, while inhibiting phosphorylation of nuclear factor kappa B (NFκB) and inducible nitric oxide synthase (iNOS) expression in CS-induced COPD. In summary, our study reveals a protective effect and putative mechanism of TA action involving the inhibition of inflammation by regulating AMPK/Nrf2 and NFκB pathways. Our findings suggest that TAs could be considered as a promising functional material for treating CS-induced COPD.

## 1. Introduction

Persistent inflammation symptoms and the irreversible limitation of airflow are considered characteristics of chronic obstructive pulmonary disease (COPD). Exposure to noxious particles or gases seems to be the major cause of the disease [1]. COPD has become a serious health problem and is the third leading cause of death in the world [2].

Inflammation is considered to be the essential factor in the pathogenesis of COPD [3]. Previous studies demonstrated that the prevalence of COPD has a strong association with cigarette smoking (CS) [4,5]. As the major source of reactive oxygen species (ROS) in COPD patients, CS could directly induce a chronic and abnormal pulmonary inflammation response in the airways of the lungs [3,6]. Growing evidence suggests that lung inflammatory responses and injury are modulated by AMP-activated protein kinase (AMPK) activation [7,8]; the effect is probably regulated by a master transcription factor at downstream nuclear factor erythroid-2-related factor-2 (Nrf2), as regulated Nrf2 can inhibit oxidative stress [9]. CS-induced inflammatory mediator release could cause the airway inflammatory response in COPD, including interleukin 6 (IL-6), IL-1β, IL-2, tumor necrosis factor α (TNF-α), and inducible nitric oxide synthase (iNOS) [10,11,12,13]. It is well documented that, as a member of the transcription factors, nuclear factor kappa B (NFκB) is involved in the regulation of the inflammatory mediator and immune regulatory pathways mentioned above [13,14].

Nowadays, the recommended pharmacotherapy for COPD can only improve symptoms, but there is no effective therapy to modulate airway inflammation in the progression of CS-induced COPD [15]. Consequently, there is an urgent need for the development of new types of anti-inflammatory medicines for CS-induced COPD. Loquat (*Eriobotrya japonica*) is an Asian evergreen tree cultivated for its fruit. Its leaves have been used for centuries to treat the symptoms and signs of COPD in clinical use [16]. Moreover, loquat leaf is also an important material for food application. In China and Japan, loquat leaf has been used as the principal ingredient of some healthy and tasty herbal teas [17], and has been processed into delicious jams and candies [18]. It is believed that eating these processed food products of loquat leaf can have similar pharmacological effects [16]. Triterpene acids (TAs) from loquat leaf were proven to be an effective component due to their remarkable anti-inflammatory effects [19,20,21]. It was shown that TAs significantly decreased the production of inflammatory cytokine in a chronic bronchitis model induced by the endotracheal instillation of lipopolysaccharide [22] via inhibition of NF-κB activation. However, there is no study available about the effect of TAs on inflammation in CS-induced COPD.

The purpose of this study is to investigate the pharmacological activity of TAs in COPD inflammation under CS exposure and explore the underlying action of the molecular mechanism. The data in our experiment present credible evidence for TAs to treat inflammation in CS-induced COPD.

## 2. Materials and Methods 

### 2.1. Plant Materials

The raw materials of loquat leaf were collected from Suzhou, Jiangsu Province, China, in October 2016. Loquat leaf was identified by the corresponding authors. The voucher specimen (No. 328636) was stored at the herbarium of our institute.

### 2.2. Chemicals, Reagents, and Antibodies

High-performance liquid chromatography (HPLC)-grade methanol and ammonium acetate were obtained from Fisher Scientific (Schwerte, Germany). A Milli-Q system (Millipore, Bedford, MA, USA) was used to generate HPLC-grade water. Tormentic acid, corosolic acid, maslinic acid, and euscaphic acid were prepared in our laboratory following a previously described method (purity > 95%) [23]. Oleanolic acid and ursolic acid were purchased from Beijing National Institutes for Food and Drug Control of China (China).

Commercial kits for quantifying superoxide dismutase (SOD) and malondialdehyde (MDA) were purchased from Nanjing Jiancheng Bioengineering Institute (Nanjing, China). Enzyme-linked immunosorbent assay (ELISA) kits of mouse IL-1β, IL-2, IL-6, and TNF-α were purchased from MultiSciences (LiankeBio, Hangzhou, China). 

The following primary antibodies and dilutions for Western blotting (WB) were used: AMPKα (1:1000), p-AMPKα (Thr172) (1:1000), Nrf2 (1:1000), NFκB (1:1000), p-NFκB (1:1000), iNOS (1:1000), β-actin (1:1000), and GAPDH (1:1000), obtained from Cell Signaling Technology (Danvers, MA, USA). Horseradish peroxide (HRP)-conjugated secondary antibodies diluted to 1:3000 were purchased from Cell Signaling Technology (Danvers, MA, USA).

### 2.3. Isolation of TA from Loquat Leaf 

TA was extracted from loquat leaf by a previously described method with some minor alterations according to the study of Wu et al. [24]. In short, dried powder (1000 g) of loquat leaf was extracted with 95% ethanol (W:V (g:mL) = 1:5) twice (2 h each time) in a 90 °C water bath. The ethanol extract was filtered and concentrated. Then, the extract was passed through an activated charcoal column (4 × 50 cm) eluted with 95% ethanol. The eluted fraction was collected, enriched, and lyophilized to obtain triterpene acid (TA; white powder). The final yield of TA isolated from loquat leaf was 5.8%.

### 2.4. Chromatographic Conditions of HPLC

TA was dissolved in methanol (1.0 mg/mL) and injected into a Dionex Ultimate 3000 HPLC system, which was equipped with an Alltech 3300 evaporative light-scattering detector (ELSD) for HPLC analysis. A Dionex Acclaim C18 column from Thermo Fisher (4.6 × 250 mm, 5 μm) was applied for analysis. The mobile phase was composed of methanol and 0.5% ammonium acetate aqueous solution (67:21) under isocratic elution conditions. The flow rate was 1.0 mL/min. The ELSD drift tube temperature was maintained at 70 °C and 1.5 L/min was set as the flow rate of nitrogen.

### 2.5. Animals

The Animal Ethics Committee of China Pharmaceutical University (certificate number: SYXK2016-0011) authorized the animal experiments in this study. Specific pathogen-free C57BL/6J mice (male, 20–25 g) from 6 to 8 weeks old were obtained from Shanghai Sino-British SIPPR/BK Lab Animal Co., Ltd. (China). Groups of 10 mice were kept in cages under controlled conditions: temperature (21 ± 2 °C), relative humidity (60% ± 5%), light/dark cycle (12/12 h), and they had free access to food and water. 

### 2.6. Induction of COPD in Mice with CS and Drug Administration

Forty C57BL/6 mice were randomly assigned to four groups (10 animals each): control group (CON), CS model group (CS), CS + 50 mg/kg TA group, and CS + 100 mg/kg TA group. During each CS exposure, except the control group, all mice were placed into the exposure chamber (40 × 25 × 20 cm). CS was generated from commercial filtered Hongta cigarettes from Hongta Tobacco Group Company Limited (Yuxi, China), containing 10 mg carbon monoxide, 10 mg tar, and 0.9 mg nicotine per cigarette. To mimic the smoking inhalation quantity and rate of combustion, cigarettes were coupled to a plastic 50 mL syringe to draw and inject puffs into the exposure chamber. According to previous studies [25,26], the procedure of passive smoking inducing COPD was conducted as follows: fresh smoke generated from four cigarettes was delivered to the chamber four times a day, twice in the morning (beginning at 09:00) and twice in the afternoon (beginning at 15:00). The smoke-free interval between two exposures was 10 min. The mice in the CS group were treated with CS exposure for 12 weeks (5 days per week). The control group was exposed to fresh air instead. From the seventh week of CS exposure, in the TA group, 50 mg/kg or 100 mg/kg TA was intragastrically administered to mice 1 h before CS exposure for 6 weeks. Mice in the control and CS model groups were administered with equal volumes of saline.

### 2.7. Lung Histological Analysis

Part of the lung was fixed in 4% paraformaldehyde, and then embedded in paraffin for histological examination. The tissues were cut into sections 4 μm thick and then stained with hematoxylin and eosin (H&E). The mean linear intercept (MLI) and destructive index (DI) were determined in lung tissues according to a reported method [27]. MLI was used to indicate the average size of alveoli. DI, which represents the percentage of destroyed space as a fraction of the total alveolar and duct space, was used to estimate the damage to alveolar walls. 

### 2.8. Quantitation of Inflammatory Mediators

IL-1β, IL-2 IL-6, and TNF-α levels in the serum were detected by commercial ELISA kits. The operating procedures strictly followed the manufacturer’s instructions.

### 2.9. Measurement of SOD and MDA

Malondialdehyde (MDA) and superoxide dismutase (SOD) kits were used to determine the MDA and SOD levels according to the manufacturer’s instructions.

### 2.10. Western Blot Analysis

Lung tissues were collected on ice. Total protein was extracted, and the bicinchoninic acid (BCA) method was used to detect the protein concentration. For the analysis, 20 μg of protein was separated on 10% sodium dodecyl sulfate polyacrylamide gel electrophoresis (SDS-PAGE) and transferred to polyvinylidene difluoride (PVDF) membranes (Millipore, Bedford, MA, USA). Then, 5% skim milk was applied to block the membranes; after that, the PVDF membranes were incubated with respective primary antibodies for 2 h. Secondary antibodies and membranes were co-incubated for 2 h, and enhanced chemiluminescence (ECL; Santa Cruz Biotechnology, Santa Cruz, CA, USA) detection was used to notice the protein bands. 

### 2.11. Statistical Analysis

The results are presented as mean values ± standard error (SE) of three separate experiments from each group. GraphPad Prism (version 7.02; GraphPad Software, San Diego, CA, USA) was used to evaluate the data, and one-way ANOVA with Tukey’s multiple comparison test was used. A *p*-value less than 0.05 was considered statistically significant.

## 3. Results

### 3.1. HPLC Profile and Content Analysis of TA

A HPLC-ELSD assay was performed to figure out the probable bioactive ingredients in TAs of loquat leaf. As shown in Figure 1A, the chromatograms revealed that six triterpene acids were the main components, and all of them were quantified through reference compounds. Figure 1B shows their chemical structures, and the substance content of the six triterpene acids is shown in Table 1.

### 3.2. TA prevents Weight Loss and Pulmonary Swelling in CS-Induced COPD

Mice were exposed to CS to induce COPD and then administered different doses of TA for 6 weeks while regular CS exposure continued. CS exposure decreased the body weight of mice and significantly increased the lung index, indicating pulmonary swelling (Figure 2A,B; *p* < 0.05): Lung index (%) = lung weight (mg)/body weight (g) × 100. During the TA administration period, the decrease in body weight in COPD mice was inhibited, while the increase in lung index was reversed. Besides, treatment with TA at a higher dose did not affect the body weight and lung index in comparison with mice in the control group, which were exposed to fresh air (Figure 2C,D; *p* > 0.05).

### 3.3. TA Attenuates CS-Induced Lung Injury

As presented in Figure 3A, showing H&E staining in the mice of the control group, the alveolar structure of the lung was complete and uniform, with rare inflammatory symptoms observed. In contrast, COPD mice displayed morphological damage and inflammatory alterations, including narrowed alveolar space, thickened alveolar wall, pulmonary edema, lung congestion, and infiltration of inflammatory cells (Figure 3B). Treatment with TA improved lung histopathological damage in CS-induced COPD in a dose-dependent manner (Figure 3C,D). The MLI is widely used to indicate the average size of alveoli, and the DI value is used to estimate the damage of alveolar walls. In contrast to the control group, the MLI and DI values were found to be clearly increased in the COPD model. As shown in Figure 3E,F, TA supplementation for 6 weeks inhibited the CS-induced increase in both MLI and DI values in a dose-dependent manner (*p* < 0.05). 

### 3.4. TA Decreases COPD Inflammatory Cytokine Concentration in CS-exposed mice

As shown in Figure 4, ELISA was used to detect inflammatory cytokines to assess the degree of inflammatory reaction, including IL-1β, IL-2, IL-6, and TNF-α. In the COPD group, CS exposure dramatically increased the levels of IL-1β, IL-2, IL-6, and TNF-α in comparison to the control group (Figure 4; *p* < 0.05). TA supplementation decreased these parameters in a dose-dependent manner (Figure 4; *p* < 0.05).

### 3.5. TA Improves the Oxidative Stress Imbalance of CS-induced COPD

Oxidative stress parameters were also detected. The level of MDA in serum increased, while the activity of SOD decreased in CS-exposed COPD mice (Figure 5; *p* < 0.05). TA treatment could adjust the trend in a dose-dependent manner, especially at a higher dose of 100 mg/kg/d.

### 3.6. Expression of AMPK, Nrf2, iNOS, and NFκB in Lung Tissues of COPD Mice

As displayed in Figure 6, CS challenge inhibited the phosphorylation of AMPK and induced a reduction of Nrf2 expression level in lung tissues, while TA administration restored AMPK phosphorylation and Nrf2 expression (Figure 6A,B; *p* < 0.05). Furthermore, CS exposure induced an increase in iNOS expression and promoted the phosphorylation of NFκB. It was shown that TA supplementation could turn back these growing trends (Figure 6C,D; *p* < 0.05). 

## 4. Discussion

The present study shows that dietary TA ameliorates CS-induced COPD inflammation by inhibiting inflammatory cytokine release, pathological lung changes (lung injury), and oxidative stress in a dose-dependent manner, and the underlying mechanism of TA treatment for COPD might be mediated by modulation of the AMPK/Nrf2 and NFκB signaling pathways (Figure 7). 

It is well documented that cigarette smoking (CS) is the leading cause of the pathogenesis of COPD [28,29]. Presenting not only a progressive decline in respiratory function, COPD patients also exhibit exaggerated inflammatory responses. To date, although the mechanism of COPD has not been fully understood, chronic inflammation induced by CS is presumed to be the underlying mechanism [30,31]. As shown in previous studies, AMP-activated protein kinase (AMPK) is an essential integrator of signals that are responsible for regulating cell growth, autophagy, and metabolism [32]. Studies also show that AMPK participates in the modulation of lung inflammatory responses induced by CS [7,33,34]. Pro-inflammatory mediators such as IL-6 and IL-8 could be released in the presence of cigarette smoke extract (CSE), and treatment with a specific AMPK activator could significantly reduce pro-inflammatory cytokine release, implying that AMPK activation is helpful to inhibit the development of COPD inflammation [33]. This effect is probably modulated by nuclear factor erythroid-2-related factor-2 (Nrf2) [9]. Nrf2 modulation can intervene in cellular damage, which is essential to various types of injury in the cell. In the LPS-challenged inflammatory response model, Nrf2 activation exhibits anti-inflammatory activity, which is associated with regulation of the AMPK/Nrf2 axis [35,36]. In our study, CS exposure induced lung inflammatory response and, as a result, pro-inflammatory cytokines including IL-6, IL-1β, IL-2, and TNF-α (Figure 4) were significantly released, while phosphorylation of AMPK was inhibited and the Nrf2 expression level in lung tissues was reduced. Our research found that TA supplementation effectively restored AMPK phosphorylation and Nrf2 expression. It should be noted that the parameters investigated in the present study were serum indicators, which only represent systemic parameters. We had considered the protein expression in the lung and tried to investigate the lung-specific changes. However, we did not investigate relevant cytokine levels in bronchoalveolar lavage (BAL) fluid or in lung tissue, which could reflect the condition of lung tissue more directly. Thus, further investigation is necessary. In the CS-induced COPD model, SOD and MDA are two important parameters to estimate oxidative stress level, which has a strong association with inflammation [37]. As shown in our study, CS exposure induced an increase in MDA level and a reduction in SOD activity (Figure 5), while TA dose-dependently inhibited oxidative stress, such as upregulation of SOD activity and downregulation of MDA level. 

In addition, lung inflammatory response is also closely related to nuclear factor kappa B (NFκB). As part of a family of transcription factors, NFκB increased significantly in CS-induced COPD mice. Therefore, downregulating pro-inflammatory factors via modulation of the NFκB pathway exhibited therapeutic effects on COPD inflammation [38,39,40]. As a pro-inflammatory cytokine, prolonged inducible nitric oxide synthase (iNOS) expression is implicated in the pathophysiology of inflammatory diseases, such as pulmonary inflammation and COPD [41,42]. In CS-induced COPD inflammation, the expression of iNOS notably increased [43,44], which was involved in the activation of NFκB. Anti-inflammatory activity in CS-induced COPD was closely associated with suppression of the NFκB pathway, which resulted in a reduction in iNOS expression [13,45,46]. In the present study, TA treatment could effectively suppress elevated NFκB/iNOS in mice with CS-induced COPD inflammation.

## 5. Conclusions

The present study shows that triterpene acids (TAs) separated from loquat leaf suppress the production of inflammatory mediators in CS-induced COPD mice in a dose-dependent manner. We report the protective effect and putative mechanism of action of TAs for the first time. The therapeutic effects of TAs on CS-induced COPD inflammation may be associated with the modulation of CS-mediated AMPK/Nrf2 and NFκB/iNOS signaling pathways. This suggests that TAs could be considered as promising functional materials for lung diseases such as CS-induced COPD. 

## Figures and Tables

**Figure 1 nutrients-12-00657-f001:**
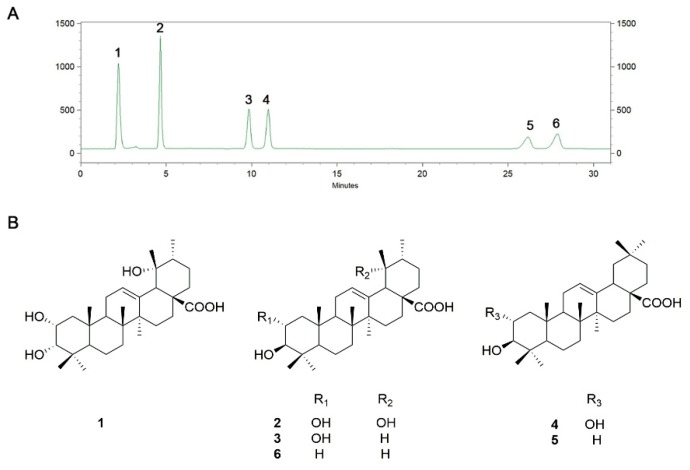
(**A**) Representative chromatograms of triterpene acids (TAs) and (**B**) chemical structures of six dominant TAs.

**Figure 2 nutrients-12-00657-f002:**
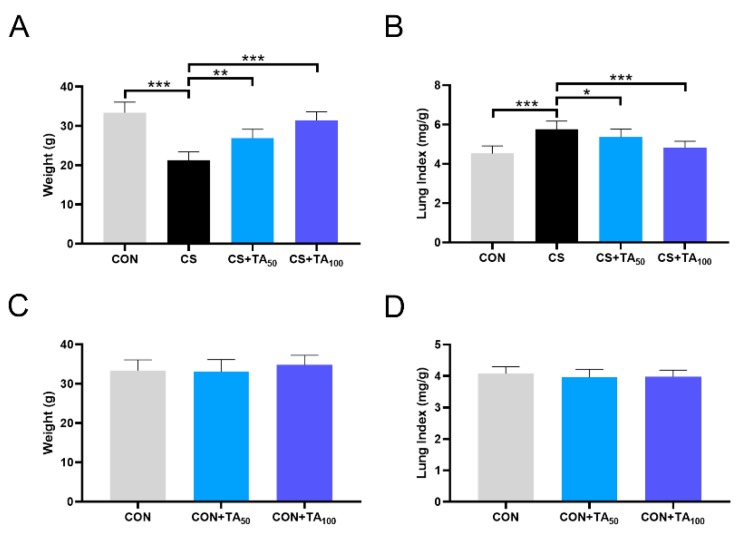
Effects of TA on (**A**) body weight and (**B**) lung index in normal mice (CON) and cigarette smoking (CS)-exposed mice treated with 50 and 100 mg/kg TA. Effects of TA on (**C**) body weight and (**D**) lung index in normal mice exposed to fresh air. * *p* < 0.05, ** *p* < 0.01, *** *p* < 0.001.

**Figure 3 nutrients-12-00657-f003:**
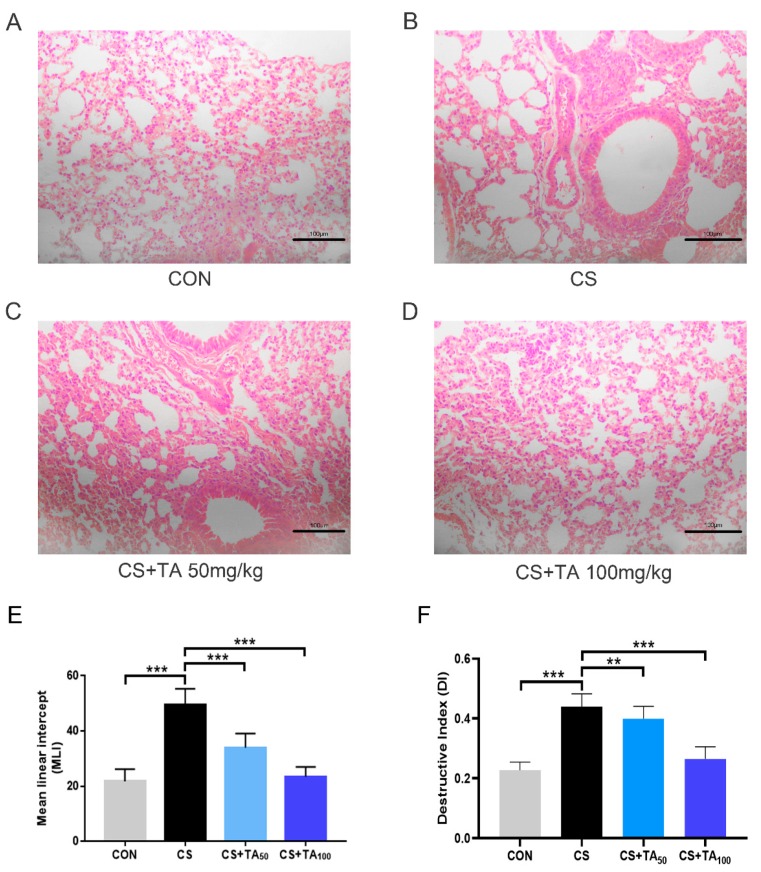
Effects of TA therapy on CS-induced pathological lung change. (**A**–**D**) Representative histopathological images (bar = 100 μm) from different groups. (**E**) Mean linear intercept (MLI) and (**F**) destructive index (DI). ** *p* <0.01, *** *p* < 0.001.

**Figure 4 nutrients-12-00657-f004:**
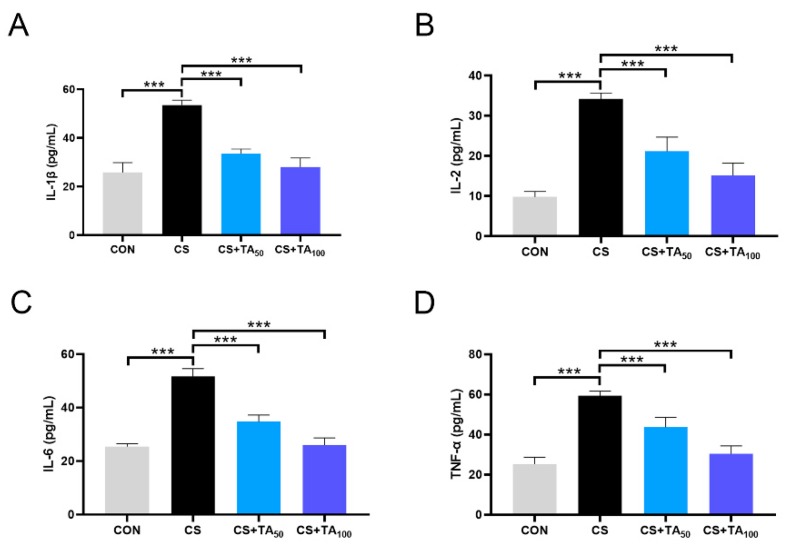
TA protects against inflammatory responses in CS-exposed COPD mice. Levels of (**A**) interleukin 1β (IL-1β), (**B**) IL-2, (**C**) IL-6, and (**D**) tumor necrosis factor alphs (TNF-α) in the serum were measured. *** *p* < 0.001.

**Figure 5 nutrients-12-00657-f005:**
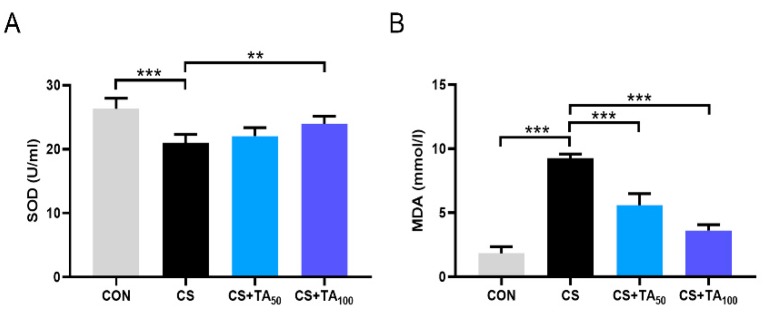
TA protects against oxidative stress in CS-exposed COPD mice. (**A**) Superoxide dismutase (SOD) activity and (**B**) malondialdehyde (MDA) levels in the serum were measured. ** *p* < 0.01, *** *p* < 0.001.

**Figure 6 nutrients-12-00657-f006:**
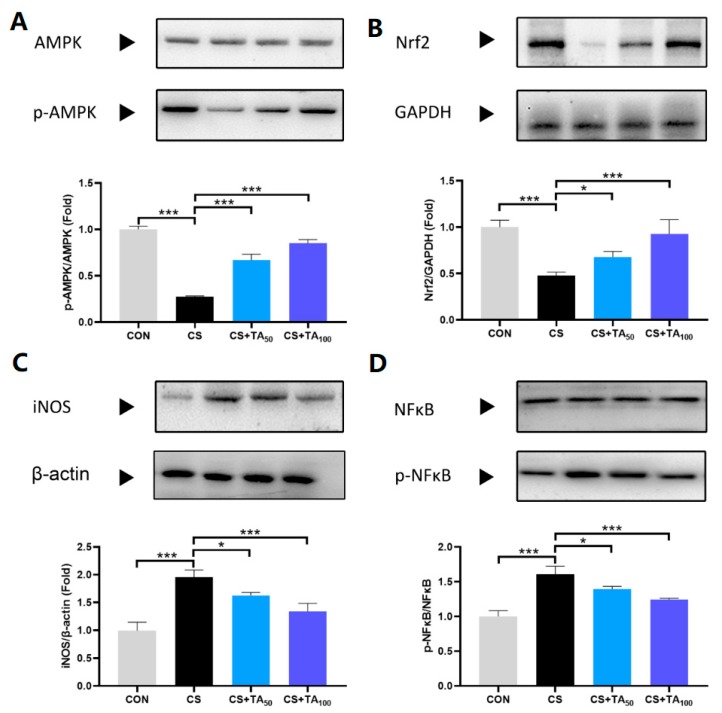
Effects of TA supplementation on AMPK and NFκB signaling pathways in the lungs of CS-exposed induced-COPD mice. Expression of (**A**) AMPK and p-AMPK, (**B**) Nrf2, (**C**) iNOS, and (**D**) NFκB and p-NFκB in lung tissues from each group were measured by Western blot. * *p* < 0.05, *** *p* < 0.001.

**Figure 7 nutrients-12-00657-f007:**
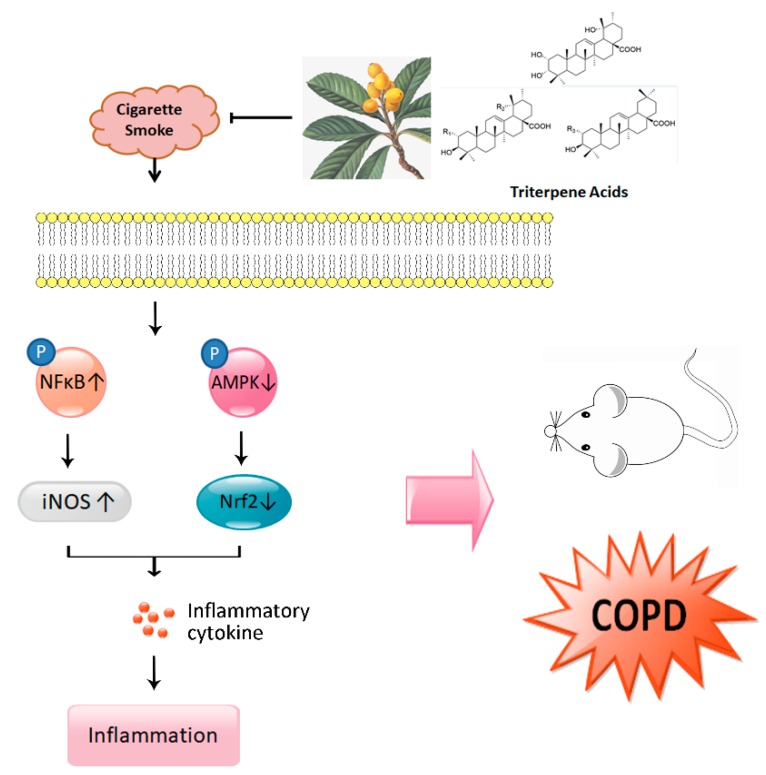
Schematic illustration of major points of conclusion.

**Table 1 nutrients-12-00657-t001:** Concentrations of six TAs analyzed by high-performance liquid chromatography–evaporative light scattering detection (HPLC-ELSD) (n = 3).

Peak Number	Compound Name	Concentration (mg/g)
1	Euscaphic acid	105.69 ± 2.10
2	Tormentic acid	155.43 ± 0.07
3	Corosolic acid	129.84 ± 6.26
4	Maslinic acid	135.24 ± 7.33
5	Oleanolic acid	65.44 ± 1.87
6	Ursolic acid	80.70 ± 0.47

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
