# Peer review of "Triterpene Acids of Loquat Leaf Improve Inflammation in Cigarette Smoking Induced COPD by Regulating AMPK/Nrf2 and NFκB Pathways"

_nutrients, 2020, doi:10.3390/nu12030657_

Round 1

Reviewer 1 Report

In this study the authors are showing Triterpene Acids can improve the inflammation in cigarette smoking induced COPD and the mechanism is through the regulation of AMPK-Nrf2 and NFκB. The manuscript is well written.

1) Fig. 4 and Fig. 5: Need to include TA treatment alone in all experiments.

2) Fig. 7: Include the increase and decrease status for pNFkb/iNOS and pAMPK/Nrf2.

Author Response

Reviewer #1

Comments

  1. Fig. 4 and Fig. 5: Need to include TA treatment alone in all experiments.

Reply: Thank you for your careful review. We understand that TA treatment alone in all experiments could better reveal the effect of TA in normal mice. In the present study, we mainly focused on the therapeutic effect of TA in COPD mice, and we believed that Fig 2C and 2D (TA treatment alone in normal mice on body weight and lung index) might not be optimal, but should be sufficient to draw a conclusion that TA treatment had no effect on the normal mice throughout the experiment. We appreciate your valuable suggestions, the referee’s concern is of importance for our further study, and we will include the results of TA treatment alone in all experiments.

  1. Fig. 7: Include the increase and decrease status for pNFkb/iNOS and pAMPK/Nrf2.

Reply: Thank you for your valuable suggestions. We have added the increase and decrease status of NFkb/iNOS and pAMPK/Nrf2 in Fig 7. We have also revised the corresponding description in the Results section of the manuscript.

Reviewer 2 Report

Jian and colleagues investigate the anti-inflammatory potential of triterpene acids (TA) from Loquat leaf in mouse COPD models. TA prevented smoke-induced inflammation and malondialdehyde, while increasing SOD, AMPK phosphorylation and Nrf2 levels. TA also inhibited phosphorylation of NFκB and iNOS expression. This is an important topic. However, there are several issues that need to be addressed. The major issues to be addressed are itemized below, not in order of significance, but in order of presentation.

Lung index measurements are not described in the methods section. What is the lung index outlined here? Equally, the histology section in the methods are limited and required further details In figure 3, please include scale bars and images from similar regions of the lungs without shadows on the images, especially regions without airways. The H&E images are of very poor quality. Destructive index measurements are included but it would be beneficial if mean linear intercepts are investigated Serum levels of cytokines were investigated but no BAL fluid levels or lung tissue levels were investigated. Please include or discuss lung-specific changes. Equally, why were SOD and MDA serum levels analyzed rather than lung-specific changes? Are these changes observed in human serum? Please discuss The manuscript requires editing by a native English speaker

Author Response

Reviewer #2

Comments

  1. Lung index measurements are not described in the methods section. What is the lung index outlined here? Equally, the histology section in the methods are limited and required further details.

Reply: Thank you for your careful review. Lung index (%) = Lung weight (mg) / Body weight (g) × 100. We have added the description of Lung index measurements and other details in the Methods section of the revised manuscript.

  1. In figure 3, please include scale bars and images from similar regions of the lungs without shadows on the images, especially regions without airways. The H&E images are of very poor quality.

Reply: As for the referee’s concern, we have added scale bars in the Figure 3. The images were re-processed of higher resolution for good quality. The Figure 3 in the Results has been re-combined in the revised manuscript.

  1. Destructive index measurements are included but it would be beneficial if mean linear intercepts are investigated.

Reply: The comment is valuable and very helpful for revising and improving our paper. According to your suggestion, we have investigated mean linear intercepts (MLI), as displayed in Figure 3E. The Figure 3 in the Results of the revised manuscript has been re-combined.

  1. Serum levels of cytokines were investigated but no BAL fluid levels or lung tissue levels were investigated. Please include or discuss lung-specific changes.

Reply: Thank you for your valuable suggestions. Serum levels of cytokines represents systemic inflammatory response induced by cigarette smoking. However, inflammatory cytokine levels in BAL fluid reflect the inflammation of the lung tissue more directly. Therefore, the referee’s concern is important but it is not available in the present study. As lung-specific changes, AMPK, Nrf2, NFκB and iNOS expression in the lung tissue were analyzed. And we have added some discussion about the lung-specific changes in the manuscript.

  1. Equally, why were SOD and MDA serum levels analyzed rather than lung-specific changes? Are these changes observed in human serum? Please discuss.

Reply: In cigarette smoking induced in COPD, oxidative stress has a closely association with inflammation [1,2]. SOD and MDA are two important parameters which could indicate oxidative stress status. In human patient with CS induced COPD, their SOD and MDA levels in the serum increased significantly [3,4]. According to your valuable suggestion, we have added some discussion about the SOD and MDA in the revised manuscript.

[1] Fischer, Bernard M., Elizabeth Pavlisko, and Judith A. Voynow. "Pathogenic triad in COPD: oxidative stress, protease–antiprotease imbalance, and inflammation." International journal of chronic obstructive pulmonary disease 6 (2011): 413.

[2] Rahman, I., and I. M. Adcock. "Oxidative stress and redox regulation of lung inflammation in COPD." European respiratory journal 28.1 (2006): 219-242.

[3] Kırkıl, Gamze, et al. "Antioxidant effect of zinc picolinate in patients with chronic obstructive pulmonary disease." Respiratory medicine 102 (2008): 840-844.

[4] Hanta, Ismail, et al. "Oxidant–antioxidant balance in patients with COPD." Lung 184.2 (2006): 51-55.

  1. The manuscript requires editing by a native English speaker.

Reply: We are very sorry for the grammar mistakes in this manuscript and the inconvenience they caused in your reading. The manuscript has been thoroughly revised and edited by a native speaker, so we hope it can meet the journal’s standard. Thanks so much for your useful comments.

Round 2

Reviewer 2 Report

Several of the concerns were addressed. However, the article is still poorly written and the images are of poor quality. Additional changes are required as outlined in the first submission

Author Response

Reviewer #2

Comments

  1. Several of the concerns were addressed. However, the article is still poorly written and the images are of poor quality. Additional changes are required as outlined in the first submission

Reply: Thank you for your careful review. We have used English Editing Services from MDPI to improve our English written, so we hope it can meet the journal’s standard. We have tried our best to process the image for higher resolution of good quality in our revised manuscript. Because of aging microscope lens, the quality of pictures remains to be improved, in future experiments, we will use the updated equipment to obtain high-quality images.
